# Mitigation Measures to Control the Expected Mpox Outbreak in a Developing Country—Pakistani Scenario

**DOI:** 10.3390/vaccines11030502

**Published:** 2023-02-21

**Authors:** Shiza Malik, Muhammad Asghar, Yasir Waheed

**Affiliations:** 1Bridging Health Foundation, Rawalpindi 46000, Pakistan; 2Department of Biology, Lund University, 22100 Lund, Sweden; 3Department of Health Biotechnology, Atta-Ur-Rahman School of Applied Biosciences (ASAB), National University of Sciences and Technology (NUST), H-12, Islamabad 44000, Pakistan; 4Office of Research, Innovation, and Commercialization (ORIC), Shaheed Zulfiqar, Ali Bhutto Medical University, Islamabad 44000, Pakistan; 5Gilbert and Rose-Marie Chagoury School of Medicine, Lebanese American University, Byblos 1401, Lebanon

**Keywords:** Mpox virus, recent outbreaks, healthcare management, mitigation measures clinical management

## Abstract

Mpox (previously named Monkeypox) is one of the neglected viral infectious diseases that remained silent for a long period before finally emerging as a threat to the healthcare system in endemic regions of the world in recent years. It has been mostly centered in African countries but has now been reported in other non-endemic regions as well. While keeping a strict eye on COVID pandemic handling, there is a need to remain concerned and alert about viral threats such as Mpox infections in the future. This situation has altered the healthcare system of endemic regions, including Pakistan, to stay vigilant against the expected Mpox outbreaks in the coming months. Though no specific cases have been reported in Pakistan, the healthcare system needs to take mitigation measures to tackle an expected threat before it arrives. This is important in order to avoid another major shock to the health care system of Pakistan. Moreover, since no specific treatment is available for Mpox, we can only rely upon mitigation measures, involving preventive and treatment strategies devised around some already in-use antiviral agents against Mpox viruses. Moreover, there is an imperative need to proactively prepare the healthcare system against Mpox outbreaks, spread awareness, and involve the public in a participatory approach to stay well prepared against any such infection. Moreover, there is a need to utilize financial sources, aids, and funds wisely, to create awareness in the public about such expected healthcare outbreaks in the future.

## 1. Introduction

With every coming decade, a new viral infection becomes the talk of the public and a threat to the healthcare system worldwide. Novel zoonotic infections continuously spread, owing to the prevailing globalization scenario worldwide, where humans are intervening in natural habitats more rigorously than ever before, and in doing so, come into contact with infectious agents in various ways through animal species [1]. Coronavirus, smallpox, cowpox, and Mpox viruses are some examples of zoonotic viral diseases [2]. Mpox infection is an emerging zoonotic disease that is mainly caused by orthopoxviruses that belong to the same family as the smallpox virus; poxviridiae. The disease symptoms are also quite similar to smallpox, which was reported to be completely removed from the world back in the 1980s with the help of a large-scale vaccination drive [3]. However, a similar infection in the form of Mpox is now threatening the world by spreading outside endemic countries. The five countries most affected by the current outbreak of Mpox are USA, Brazil, Spain, France, and Colombia [4]. 

Mpox virus is a species of large double-stranded DNA viruses, belonging to the genus Orthopoxvirus and the family of Poxviridae. Mpox disease was first reported as being of zoonotic origin in DRC back in 1970 [5]. It mostly remains endemic to 11 African countries, but mainly to DRC for the last 50 years [3,6,7]. The symptoms of Mpox infection include fever, headache, muscular pain and back aches, shivering, and extreme tiredness [8,9,10,11,12]. The lymph nodes in the ear, jaw, neck, and groin regions also noticeably swell, which is followed by rashes in the form of blisters and crusts over different exposed skin portions of the body, such as hands, feet, face, genitals, mouth, and eyes, etc. [13,14]. The symptomatic indications and disease cycles of 5–21 days are quite similar to smallpox infection, from which the world has been immune for quite some time now. The vulnerable populations include children, pregnant women, and the elderly with a suppressed immune system [14,15,16]. Owing to its similarity with smallpox infection, the smallpox vaccine can be utilized against Mpox as directed by WHO. A few vaccine candidates, including mainly the smallpox vaccine, along with therapeutics such as tecovirimat (antiviral agent), are currently being employed for the treatment of Mpox infection in endemic regions [17,18,19,20].

The close physical proximity of infected individuals may play a crucial role in virus transmission (Pastula, 2022). There are multiple routes by which Mpox virus enters host cells, such as direct contact, bodily fluids, shedding through feces, intramuscular or subcutaneous entry, mucosal surfaces, and intradermal entry. Micropinocytosis, endocytosis, and fusion processes are used for entry into host cells with the subsequent replication at the site of infection (Dai, 2022). Following this, the host activates immune responses in the form of phagocytosis or necrosis. The virus enters the lymphatic system from the bloodstream and affects other vital organs such as lymph nodes, spleens, tonsils, and bone marrow. Eventually, it multiplies and manifests itself as Mpox virus-related disease in the host (Tiecco, 2022). 

Mpox was first reported at the beginning of the 21st century and remained centered in a few countries of the African continent. However, as of 20 December 2022, WHO has estimated 82,889 confirmed cases reported from 110 countries and territories, with 65 deaths recorded [3,4,21]. The world is afraid of what has already been faced in the form of the coronavirus pandemic and this scenario dealing with another zoonotic endemic. The threat is now hovering over South Asia where some reported cases have been found in India [22]. This situation is thus threatening for Pakistan, since we are not prepared to face yet another pandemic this decade. Keeping this scenario in mind, healthcare officials through the forum of the World Health Organization (WHO) have released advisory notes and recommendation regarding the surge of Mpox infections and relatable treatment and preventive measures [23]. The top healthcare authority of Pakistan, the National Institute of Health (NIH), which is associated with different hospitals and educational platforms has also issued alerts to provincial and national healthcare bodies to direct them in the surveillance of the outbreak [24].

Unlike the coronavirus pandemic, Mpox infection has limited or no links to travel, and the endemic regions show it is related to homosexual physical interactions. Moreover, in endemic regions, the smallpox vaccine is being utilized extensively, due to its reportedly high efficiency, and it’s links with the same virus family as poxviridae [25]. Additionally, diagnostic testing services are fixed in endemic regions to impede the spread of the virus. In Pakistan, we are still fighting to enhance COVID vaccination and control the remaining COVID infections. Similarly, many other infectious diseases, such as dengue, influenza, and hepatitis, among others are burdening the healthcare system of Pakistan [26,27]. Additionally, the situation is creating havoc, as the smallpox vaccine has been discontinued for many years because WHO declared Pakistan smallpox-free back in 1980. This has created a gap in the form of susceptible communities to Mpox infection due to a lack of poxvirus immunity owing to the termination of the smallpox vaccination program [28]. Moreover, diagnostic testing against Mpox in Pakistan is almost unavailable; thus, the situation raises concerns about gaining a basic understanding of the Mpox infection, especially for medical workers, in order to define and curtail Mpox infections in the future [29]. The disturbed socio-political, and more specifically economic, situation in Pakistan, alongside its fragile healthcare system and meager research facilities, indicate another health and economic challenge in the near future if the proper actions are not taken in a timely manner [29,30].

### 1.1. Smallpox and Mpox Infection—Vaccination Criteria

As elaborated earlier, smallpox vaccines are being employed against Mpox infection. This is because there are various similarities between the two viruses. Both viruses are pathogenic and belong to the family of orthopoxvirus, as they share structural similarities [31]. Mpox infection in humans clinically resembles the discrete smallpox infection but unlike smallpox, Mpox is usually associated with lymphadenopathy in addition to the typical rash and exhibits milder symptomatology as compared to smallpox infection [32]. However, Mpox infections have often been reported to cause death (up to 10% of cases), unlike smallpox infections. Moreover, two subtypes of Mpox have been identified over the years of its genomic analyses; these are Mpox clade I (more pathogenic) and Mpox clade II which are seen in central and West African regions respectively [33]. An important difference between Mpox and smallpox is the low-efficiency rates of human-to-human transmission in the case of Mpox. This is the very reason for the limited transmission of Mpox and the lower number of outbreaks in non-endemic regions [34]. However, the fact should not be ignored that new studies are showing an increasing efficiency of Mpox spread in wider human populations and there is a need to take strict regulatory measures against further Mpox spread to non-endemic regions of the world. Due to the similarity in nature of the infection, smallpox vaccines (such as JYNNEOS™) that are made from weakened vaccine viruses have been approved for use against Mpox [33,34].

Human Mpox is becoming noticed for the increasing infection spread to different regions of the world. Moreover, the long-term absence of smallpox vaccination in the population has possibly made people more vulnerable to Mpox infection. The prevalent scenario has made Mpox infection acquire high transmissibility and pathogenicity, similar to the characteristic natural evolution of vaccinia virus (VACV) [35]. If the situation remains the same, it may become difficult to deal with Mpox eradication from the world. Therefore, to prevent the pathogenic outbreak and widespread epidemic of Mpox, instead of relying on smallpox and VACV-based vaccines against Mpox the scientific community should put in rigorous efforts to create a better and new generation of VACV-based life-saving vaccines against Mpox with greater efficacy against Mpox infection [34,35]. VACV-based vaccines will allow wider-scale vaccination since they do not have any pronounced species-specific pathogenic characteristics against orthopoxviruses and thus can be utilized for wider-scale immunization campaigns. Some other vaccines, such as IMVANEX, IMVAMUN, VACdelta6, OrthopoxVac, and JYNNEOS have been approved to be used against both smallpox and Mpox infections [31,32,33,34,35].

### 1.2. Post-Pandemic World at the Mercy of Mpox Virus Outbreak: Time to Worry or Not?

The world is still being hit by the coronavirus pandemic that peaked back in 2020 [26]. Though various antiviral treatments and vaccines are now in markets worldwide, the world is still facing hurdles in the provision of complete coronavirus immunity [36,37]. We have now accepted that coronavirus is here to stay, and this vision has changed our understanding and outlook for other endemic viral diseases [38]. With the fatality rates lying in the range of 3% to 10%, Mpox infection is a serious threat for healthcare worldwide [21,39]. Moreover, the still prevailing strains of COVID-19 make a person more vulnerable to Mpox infections [1]. Low income and middle-income countries, such as Pakistan, are prone to face deleterious healthcare conditions during ongoing circumstances. The authorities and educated communities are not well prepared and the public may not even have heard about such an infection in recent years [40]. As described earlier, to deal with the said scenario, WHO is supportively launching surveillance campaigns worldwide to assist communities and spread awareness regarding disease management [41]. Some of the recommendations proposed by these authorities, and the relevant measures taken for disease management, will be discussed below.

### 1.3. Mpox Virus Outbreak: A Brief Timeline and Scenario in Pakistan

The endemic nature of Mpox infection has slowly taken the form of an epidemic due to reported cases in different far-off countries, such as the recent reported outbreaks in Nigeria in 2022 [8]. Similarly, some other cases have been reported in the UK, back in April 2022 [38]. According to WHO, approximately 110 countries have already reported confirmed cases of Mpox, including UK, Spain, Portugal, Canada, Germany, Belgium, Italy, France, Netherlands, Sweden, UAE, Czech Republic, Brazil, America, and DRC [4,5,16,21,24,41,42,43,44,45,46,47,48,49].

In Pakistan specifically, two cases have been detected with rare infection incidences in the month of May 2022 [24]. These cases were not confirmed and were almost immediately rejected by the Ministry of National Health Services Regulations and Coordination, Government of Pakistan owing to a lack of confirmation. Afterward, the healthcare authorities released notifications for proper monitoring of the current situation and instructions to stay prepared and take mitigation measures for suspected cases in the future [27]. Though the immediate efforts put forth in the case of the COVID-19 disease in terms of disease management, diagnosis, and vaccination were appreciated by different world healthcare authorities, such as WHO, the efforts to protect against the Mpox virus could be a new challenge to Pakistan in the coming days [50]. This is because the smallpox vaccine has been fully terminated in Pakistan and the diagnostic facilities against Mpox are almost nonexistent and not well known [28]. This scenario has been taken into consideration by the health ministry which is trying to procure diagnostic kits for various hospitals in Pakistan [27,50].

Although the travel connection of Mpox is not reported extra vigorously, as in the case of the coronavirus, there is always a chance to spread viral infection by travel. A large diaspora of Pakistan (approximately 4% of the population; 8.8 million) lives, studies, works, and travels abroad every year according to the population estimates of 2017. The number are likely to have increased in the years 2018-till now and thus the chances of the virus spread within Pakistan are high [27,51]. This is based on past incidences of prevalence, transfer, and upbringing of endemic and epidemic, and non-communicable diseases within Pakistan that were most likely brought by traveling carriers [50]. Under the described scenario it is pertinent to take timely actions regarding awareness of the populace, establishing isolation centers, contact tracing, designing or importing diagnostic kits, approving therapeutics against Mpox, improving pre-exposure and post-exposure vaccination and vaccination re-dosages to curtail viral outbreaks in coming months [52].

The resumption of social life following the release of coronavirus restrictions can be an attributive factor for the spread of other viral infections, as can be observed with the rising incidence of dengue infections [52,53]. Thus, close physical contact, ease of social distancing, and relieving travel bans could combine to encourage viral transmission [54,55,56].

## 2. Is There a Need for Mpox Vaccination Amidst the Hesitancy of COVID-19 Immunization in Pakistan?

Vaccine hesitancy is among the top ten challenges faced by WHO which has extended after the COVID-19 pandemic worldwide [18]. With regard to Pakistan specifically, the country has always suffered from the issue of vaccination hesitancy, such as for the polio vaccination, COVID-19 vaccination, or for the proposal of the current Mpox vaccination drive [57]. Though no official Mpox cases have been reported yet, from the coronavirus pandemic we have learned one thing about viral infections; no country can rely on statistics and news to compromise their healthcare system. It only takes a couple of days before an endemic acquires the shape of an epidemic and then becomes pandemic, spreading to hundreds of countries [58].

Currently, 110 countries are affected by Mpox and it cannot be said for certain where Mpox infection is likely to stop, or how long it will take to curtail Mpox infections [42,59]. Therefore, Pakistan’s healthcare sector needs to face these challenges head-on and try to manage the disease burden, as well as dealing with financial dearth, insufficient medical staff, and nonviability of ventilators, hospital beds, isolation centers, laboratory equipment, and needful diagnostic kits against Mpox [50,58,60]. The same scenario was faced during the COVID-19 pandemic period, when there were only limited healthcare facilities available in hospitals in Pakistan, and the limited number of isolation centers and facilities of ventilators alarmed the healthcare communities with regard to the future preparedness for such pathogenic outbreaks. The scenario was even worse during the later periods of corona variants (Delta and Omicron) since no specialized diagnostic kits were then available for identification, diagnosis, and relevant treatment measures against specific cases. Therefore, more well-organized healthcare campaigns and educational and medical strategies should be implemented by the government and private sector coordination in Pakistan to disseminate accurate and needful timely mitigation measures against Mpox infection, which may significantly contribute to dealing effectively with vaccination hesitancy in Pakistan [14,51].

## 3. Recommended Mitigation Measures against Mpox for Developing Countries, Specifically Pakistan

Mitigation measures may include several recommendations underlying social awareness protocols, participatory approaches, campaigns and strategies for improvising and vaccination drives to reduce vaccine hesitancy, needful diagnostic measures, developing research strategies, and coordination and well-connected record keeping of medical data to ensure the timely detection of Mpox in Pakistan [1,61,62]. Similar steps and mitigation measures should be encouraged for other viral infections, co-epidemics, and co-occurrences such as Dengue virus disease, Zika virus infection, Chikungunya, Crimean Congo hemorrhagic fever, measles, and the still prevalent Poliomyelitis along with the COVID-19 infection all over the world, especially in Pakistan [61,62].

### 3.1. Social Awareness Program

Some important measure regarding social awareness against Mpox and other infectious viral outbreaks may include [9,10,63]:Healthcare professionals and doctors have significant roles when it comes to counteracting doubts about vaccination drives; therefore, they should play a proactive role in educating the community to actively participate in awareness programs and vaccination drives [27].The significance of good hygiene habits, clean eating and drinking habits, self-quarantining, and immediate healthcare consultancy should be encouraged as precautionary measures against Mpox-like viral disease outspread in Pakistan [64].The institutional capacities, research and medical facilities, and public response rate to healthcare awareness that have been increased during the coronavirus pandemic in Pakistan should be effectively utilized against Mpox and other viral diseases [53].General precautionary measures proposed and implemented for coronavirus should also be practiced for Mpox virus, especially in suspected areas since Mpox transmission mode is still not fully comprehended [65].Misinformation through social media platforms should be strictly dealt with to avoid concern by the public.Physicians and researchers are especially required to stay aware and to have knowledge about Mpox disease infectious symptoms to know the need for precautionary measures for patients and their workplace [66].Homosexual practices should be strictly avoided and disease spread associated with it must be talked about in public [66].Similarly, the educated youth should play a role, by coordinating with the medical industry and the general public in a participatory approach to deal with vaccine hesitancy and Mpox preventive measures to lessen the burden coupled with the COVID-19 pandemic on Pakistan [2,53].Social media platforms, print, and digital media, along with radio and TV, should be effectively utilized to spread awareness regarding likely Mpox infection along with multiple other viral infections as discussed earlier.

### 3.2. Treatment Strategies and Vaccination Strategies against Mpox

Treatment and preventive strategies are of the utmost importance in any viral outbreak, and thus the following [18] recommendations should be implemented to increase treatment and preventive practices against Mpox and related infections in developing nations, such as Pakistan [11,12,67,68].

Healthcare agencies should ensure the timely import of approved vaccines against Mpox and if possible, they should try to develop vaccines without compromising the quality.Antivirals against Mpox that have been approved (Tecovirimat, Modified Vaccinia Ankara virus, Cidofovir, and Vaccinia Immune Globulin Intravenous (VIGIV)) should be made available in a proactive way instead of waiting for confirmed cases of Mpox in Pakistan [64].Postexposure vaccination may also be performed for frontline healthcare workers against Mpox [18,68].Personalized therapies may be encouraged based on patients’ vulnerability and probability of developing Mpox infection [57].Tentatively approved smallpox vaccine and JYNNEOS vaccine should be imported in Pakistan in limited amounts to immediately deploy in case of any detected viral infection [57,68].Proper steps should be taken for disease management and treatment of patients with symptoms, coupled with nutritional and psychological support for the patients [28].It is better to vaccinate health workers before disease exposure to avoid any future risk of Mpox infection in them.

### 3.3. Diagnostic Measures

According to the estimates of health-related ministries and authorities in Pakistan, presently hospitals lack proper diagnostic facilities against Mpox and thus the following measures should be put into practice for the good management of the disease in the future [26,68].

Researchers should largely collaborate to develop specific polymerase chain reaction (PCR) protocols and diagnostic kits for molecular-specific detection of Mpox [13].Instead of sending diagnostics samples abroad and increasing the threat of viral infection spread, laboratories should be equipped with diagnostic technologies.Secondary viral, bacterial, or any pathogenic infection should be strictly diagnosed, treated, and managed well along with Mpox infection propagation in patients [10].The diagnostic symptoms, such as skin lesions, fluid-filled vesicles and pustules, and dry crusts similar to smallpox are the symptoms that should be kept in consideration by the general public to detect that they may have contracted Mpox disease [64,65,66,67].It should be explained that antigens, antibodies, and ELISA testing, do not offer Mpox confirmation because of the viral similarity to smallpox. This is an important point to be noted because patients may be diagnosed falsely positive for Mpox when they may have developed results owing to smallpox immunization [13,64,65,66,67,68].WHO has largely confirmed conventional PCR for diagnosing suspected Mpox cases. therefore it is pertinent to ensure PCR machines are in advanced laboratories in Pakistan along with the provision of required testing kits, primers, and antiviral reagents against the Mpox epidemic [17,67].

### 3.4. Research Strategies

Special laboratory conditions and safety precautions, along with SOPs, should be used in laboratory work with viral infectious diseases research. This is important to minimize the likelihood of diseases in laboratory workers [26,67].The virus study-based labs should use genome sequencing technology to detect mutation in virus clades of different zoonotic diseases that may develop into other zoonotic infections in the future. Moreover, the data delivered by WHO, CDC, and other healthcare authorities should always be kept in mind for monitoring and outlining the research plans against antiviral diseases [27].Laboratory staff should be trained in sample handling, processing and self-precautionary measures to be adopted while working in virus labs [61].Authorities should keep a strict eye and make regulations or restrictions for importing animal models for research experiments from endemic countries to avoid the zoonotic outspread of Mpox [61,62,63].Pharmaceutical companies should import and look into the approved antiviral agents for Mpox disease management and supportive care provision to suspected patients.Govt of Pakistan should collaborate with international healthcare agencies and institutions to get financial and research resources. Moreover, the Ministry of Health should effectively manage the healthcare budget in the right direction alongside raising appropriate funding for needful measures.

### 3.5. Screening, Coordination, Tracing Data, and Record Keeping for Mpox

Virus screening through PCR detection methods should be ensured at airports to avoid Mpox spread from endemic to neutral countries [61].Medical records on the baseline health status, immunization status, and co-morbidity record of patients should be easily available [24,62].Patients with suspected disease must be isolated for a prodromal period and must be immediately reported to relevant authorities. Sampling from such patients should be immediately tested in laboratories for confirmation [62].More accurate literature and research output along with disease surveillance strategies should be generated in line with epidemiological accounts, host reservoir information, transmission strategies, epigenetic factors, and genetic diversity of the disease [63,64].A global collaborative international effort in terms of contact tracing, surveillance, and rapid diagnosis, like that of the COVID-19 pandemic, should be developed for curbing Mpox and other expected viral outbreaks in the future [8].There is still a dearth of knowledge and research on various aspects of Mpox infection; therefore there is a need to carry out more relevant research studies with a focus on therapeutics and vaccination development against Mpox.Specialized personnel such as virologists, biologists, vaccinologists, public health workers, pharmacologists, and veterinary doctors should all join hands to raise public awareness in the general population, especially in areas where living conditions and some suspects have been recorded as carrying Mpox [8].Healthcare professionals, doctors and laboratory technicians should handle the suspected cases carefully because there is still unrefined information for the spread of Mpox, which makes the front-line a likely target of Mpox infection [67].Suspected cases, which are still rarely detected, should be isolated; however, the human condition should be ensured for dealing with such patients.Public should be made aware of disease symptoms and preventive and treatment measures, along with educating them on topics such as sexual health, collaboration, and communication with healthcare authorities among others [44].Professional workers who may develop disease symptoms do not need to be isolated like coronavirus patients, yet need active surveillance against disease development.Proper surveillance and record keeping are necessary for not only Mpox but for all other viral infections to know exactly where we stand and what measures should be adopted for future disease management.Contact tracing should be ensured in confirmed cases for Mpox.Healthcare communities should collaborate to design a way forward for overcoming the outbreak by planning for close gatherings, especially in education institutes, daycare centers, and healthcare centers that serve as hubs for viral outspread [62].Simple precautions, such as washing hands, keeping social distance, and avoiding unsafe sexual practices are first-line strategies to avoid any viral infection and these should be promoted in public [5].Captive animals, personal pets, and research models that show viral disease symptoms should be immediately isolated for at least 30 days. Moreover, meat, blood, or other bodily secretions should not be used or consumed from sick or dead animals. Additionally, meat and milk products should be thoroughly cooked and boiled before eating [21].If in the future there comes confirmation of Mpox disease, a public health emergency should be imposed in suspected and vulnerable areas and needful measures should be taken to monitor and manage the disease response system [68].

## 4. Conclusions

Developing countries around the world, especially Pakistan, are already burdened with the implications of the coronavirus pandemic for the past three years. The social, political, and economic implications have greatly affected the developmental pace of Pakistan. Thus it cannot afford to bear a burden more profound than coronavirus in terms of any likely future viral diseases. Therefore, to avoid such a scenario it is important to take needful mitigation measures in terms of healthcare provision, diagnostic services availability, participatory approaches for educating the public, and a proactive role and planning by healthcare authorities and professional training. By effectively implementing the needful preventive measures it is possible to avoid an Mpox infection outbreak in Pakistan in the coming years. Though it is still unreported, what is needed is to be proactive and update the expert issues guidelines, start vaccination campaigns, and enable research in a similar domain. Only by adopting these measures can we ensure Mpox diseases remain in a containable form unlike that of the coronavirus pandemic, within Pakistan and other non-endemic countries. The only sound step is to proactively tackle potential viral outbreaks.

## Data Availability

Not applicable.

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
