# Peer review of "Mitigation Measures to Control the Expected Mpox Outbreak in a Developing Country—Pakistani Scenario"

_vaccines, 2023, doi:10.3390/vaccines11030502_

Round 1
Reviewer 1 Report
I thank the academic editor for giving me the opportunity to review this manuscript in "commentary" form. I think the authors' goal of informing, treating and raising awareness of the problem focused at Pakistan is a good idea, but I ask the authors to fix a few things:
1) first of all, I ask them to write more appropriately as in some parts the manuscript is difficult to understand, with many repetitions of names and sentences for no reason.
2) I suggest correcting English language errors and typos that are present throughout the manuscript.
3) I suggest studying, citing and discussing the following paper: MDPI and ACS Style
Ambrogio, F.; Laface, C.; De Caro, A.P.; Loconsole, D.; Centrone, F.; Lettini, T.; Cazzato, G.; Bonamonte, D.; Foti, C.; Chironna, M.; Romita, P. Peri-oral Monkeypox Virus Infection: A Clinical Report with Confirmatory Polymerase Chain Reaction Findings. Vaccines 2023, 11, 36. https://doi.org/10.3390/vaccines11010036
Author Response
"Please see the attachment."

Reviewer 2 Report
Speculative materials not backed by hard scientific data

Author Response
"Please see the attachment."

Reviewer 3 Report
This paper discussed on how to avoid Mpox hitting Pakistan. The authors introduced Mpox and the public health situations in Pakistan and provided useful information on the perspectives of social awareness, treatment/vaccines, research, and diagnosis with necessary steps afterwards. Since COVID is a virus that caused the most recent pandemic in human history and is still impacting people worldwide, it is very important to get prepared for Mpox, a very critical global health threat that has already showed some potential of impacts on many regions around the world. The paper has high significance and has provided readers a lot of useful information on viral prevention. Please also check the comments below on making the paper better:
1. Abstract is a bit redundant. Please re-organize the text with a logical flow.
2. In introduction, first paragraph, you mentioned that Mpox is similar to smallpox. Then you mentioned people using smallpox vaccines to prevent Mpox. Later, on page 3, under and isolated topic, you described symptoms caused by Mpox. I would suggest you move the symptom information to introduction part, and add more details, like transmissions (since you talked about that on page 3), infection latent period, time span etc. Also, please mention the differences and similarities between Mpox and smallpox. In a conclusion, the order is introduction (which includes details on Mpox list above in this comment), time to worry or not (subtitle 1), then Pakistani scenario (subtitle 2).
3. Adding more to comment 2, please also mention the efficiency using smallpox vaccines against Mpox infections.
4. For some reason, I feel many discussions on page 2-4, are somehow superficial. For instance, line 133 to 142, you listed many things, but no details. Still using this part as an example, you can provide some demographic data on how many people live, study, travel or work abroad. When you talk about isolation centers, why isolation centers? Any centers established during COVID time? How many, and when were the centers being used (under the first wave of COVID? Or during Delta/Omicron time?)? Similar ideas also apply to diagnostic kits and therapeutics. Also, any examples on defending other viruses can be use here in parallel? Overall, you have some subtitles on page 2-4, each means a specific topic. These topics are nice, however, more details will really help a lot in making the paper comprehensive and deep.
Author Response
"Please see the attachment."

Round 2
Reviewer 1 Report
none
Reviewer 2 Report
Use practical scenarios for control of Mpox

Reviewer 3 Report
The authors have made changes based on the comments. However, for the last comment, I still think using COVID situations as an analogy example in during discussions will be nice, since COVID is a hotspot virus that has been, and still impacting the entire world, and Mpox is something that is causing a lot of tension after COVID. You don't need to have super detailed info, but again, using COVID as an analogy during discussions may help.